# Does the Great Recession Contribute to the Convergence of Health Care Expenditures in the US States?

**DOI:** 10.3390/ijerph17020554

**Published:** 2020-01-15

**Authors:** Jesús Clemente, Angelina Lázaro-Alquézar, Antonio Montañés

**Affiliations:** 1Department of Economic Analysis, University of Zaragoza, 50005 Zaragoza, Spain; clemente@unizar.es; 2Department of Applied Economics, University of Zaragoza, 50005 Zaragoza, Spain; alazaro@unizar.es

**Keywords:** great recession, health care expenditures, long-term, convergence analysis, Phillips-Sul

## Abstract

This paper examines whether the Great Recession has altered the disparities of the US regional health care expenditures. We test the null hypothesis of convergence for the US real per capita health expenditure for the period 1980–2014. Our results indicate that the null hypothesis of convergence is clearly rejected for the total sample as well as for the pre-Great Recession period. Thus, no changes are found in this regard. However, we find that the Great Recession has modified the composition of the estimated convergence clubs, offering a much more concentrated picture in 2014 than in 2008, with most of the states included in a big club, and only 5 (Nevada, Utah, Arizona, Colorado and Georgia) exhibiting a different pattern of behavior. These two estimated clubs diverge and, consequently, the disparities in the regional health sector have increased.

## 1. Introduction

Economists agree that the Great Recession has been one of the deepest and most extensive economic downturns in recent history. The health sector did not escape this effect and suffered an immediate and long-lasting impact. If we focus on the case of the USA, Martin el al. [1,2] show that US National health spending grew by 4.6% in 2008 and 3.9% in 2009. After 2009, the growth rates of health spending remained below 4% for five consecutive years, with the remarkable minimum of 2.9% in 2013. This behavior meant that the health expenditure share over GDP remained unchanged until 2014.

This decline in US health spending has been quite heterogeneous. If we analyze the Personal Health Care Expenditure (PHCE) data for 2009–2014, we observe that it grew by 6.7% in North Dakota and by 2.5%, the lowest, in Rhode Island. If we further take into account that California’s PHCE is 11.5% of US total, whereas the contribution of Wyoming is just 0.2%, we can understand that US health spending is heterogeneous across US States.

Some recent papers have analyzed these disparities, putting special emphasis on the convergence of the US regional health expenditure. For instance, Cuckler et al. [3] and, more recently, Apergis et al. [4] and Clemente et al. [5], study convergence with State of Residence health data. Similarly, Wang [6] and Panopoulou and Pantelidis [7] consider State of Provider health data. The results obtained by these authors are mixed, although evidence of convergence is scarce. Other interesting works related with this issue are the ones of Caporale et al. [8] and Zezza [9]. However, none of these previous papers deal directly with the possible effect of the Great Recession on convergence.

Against this background, the aim of this paper is to analyze whether the Great Recession has affected the disparities in US regional health care expenditures. We focus on State of Provider health data and our sample covers 1980–2014. Therefore, our results can be understood as an extension of those of Panopoulu and Pantelidis [7] for the post-Great Recession period.

The structure of the paper is as follows. Section 2 describes the data and introduces the methodology. Section 3 presents and discusses the empirical results. Section 4 offers some concluding remarks.

## 2. Materials and Methods

The annual data of Personal Health Care from 1980 to 2014 have been obtained from the Center of Medicare and Medicaid Services (CMS) for the 50 US states plus the ones of the District of Columbia. Unlike Clemente et al. [5], this variable is measured by State of Provider. We have transformed the data into real per capita terms by using the population of each state and the US consumer price index, the data for which have been obtained from the US Bureau of Economic Analysis.

To analyze the disparities in US regional health expenditures, we apply the methodology of Phillips and Sul [10,11]. They develop a procedure that allows, first, to test the total convergence hypothesis and, if this hypothesis is rejected, to subsequently estimate the number of different convergence clubs. Details of this methodology are presented in a separate Appendix A.

## 3. Results and Discussion

The results of Table 1 show that the null hypothesis of convergence is clearly rejected for PHCE for 1980–2007 and for the total sample (1980–2014). So, nothing has changed from this perspective. However, we should note that the estimation of the parameter that measures the speed of convergence has augmented in absolute terms, going from −0.58 to −0.78. Similarly, the statistic for testing the null hypothesis of convergence goes from −45.2 to −66.0. Thus, greater heterogeneity is found after the Great Recession.

Next, we analyze the existence of convergence clubs in the US States. The results are presented in Panel II of Table 1 and mapped in Figure 1. As can be seen, we have estimated two clubs. Club 1 includes the states with the largest PHCE while Club 2 includes the states with the lowest levels of PHCE. The states in Club 2 for the total sample are AZ, CO, GA, NV and UT, clearly fewer than those in Club 2 for the pre-Great Recession sample (21 states).

To study whether the differences in the composition of the clubs are statistically relevant, we have employed the Mann-Whitney-Wilcoxon, Kruskal-Wallis and van der Waarden non-parametric statistics. Their respective values are 3.61, 13.09, and 13.63, rejecting the null hypothesis of the median equity when the results of the pre-Great Recession and total sample are compared. Therefore, our results provide robust evidence against the hypothesis that the estimated convergence clubs are similar for both the pre- and the post-crisis samples. Thus, it can be concluded that the Great Recession has statistically changed the pattern of behavior of health expenditure in the US.

To appreciate the changes caused by the Great Recession, Figure 2 shows the average values of PHCE for the states in club 1 and club 2 when the total sample is considered. We can observe that their growth rates diminished after 2008. However, the states in Club 2 did so more intensively than those in Club 1. Therefore, the distance between the two clubs has augmented in the 2008–2014 period, increasing the divergence between them.

The estimated clubs presented in this paper are similar to those of Clemente et al. [7], where state-of-residence data is employed, although the final dimension of our Club 2 is clearly lower and, therefore, the effect of the Great Recession is more intensely observed. Finally, we should note that the analysis of the forces that drive the creation of the clubs should offer similar results and to the one carried out in the previously mentioned work, where the per capita GDP plays a crucial role.

## 4. Conclusions

The Great Recession has had an important impact on most sectors of the US economy, especially in health care expenditure given its dependence of the evolution of the economy. As PHCE varies considerably across states, we have analyzed whether the economic crisis has affected the convergence of health expenditure in the US states. Our empirical results reject the null hypothesis of convergence for the two samples considered. We have also found that the estimated convergence clubs have been affected by the Great Recession because the composition of the estimated clubs for the total sample is different to that of the pre-Great Recession sample. In particular, we can observe the creation of a “big club”, which includes most of the States, with only five states (Nevada, Utah, Arizona, Colorado and Georgia) in Club 2. There exists a growing gap between these two clubs, although we can appreciate a slight reduction of this divergent process in the last two years of the sample. Thus, it seems that the recovery of the economy may help to reduce the distance between the health expenditures of the two estimated clubs, though this would require further analysis when new data are available.

Finally, given that the relationship between health care spending and the quality of care is a fundamental part of understanding geographic variations, our quantitative analysis of expenditure should be complemented with a study of the efficiency of the US health system. Then, it would be interesting to extend our study to include the analysis of health outcomes (infant mortality rate or life expectancy, for instance. Similarly, and as Nghiem and Connelly [12] do for the OECD countries, the analysis of the determinants of the health care expenditures could provide additional insights. However, we have preferred to leave both analyses for future studies given that they require more complex analysis than the ones presented in this note.

## Figures and Tables

**Figure 1 ijerph-17-00554-f001:**
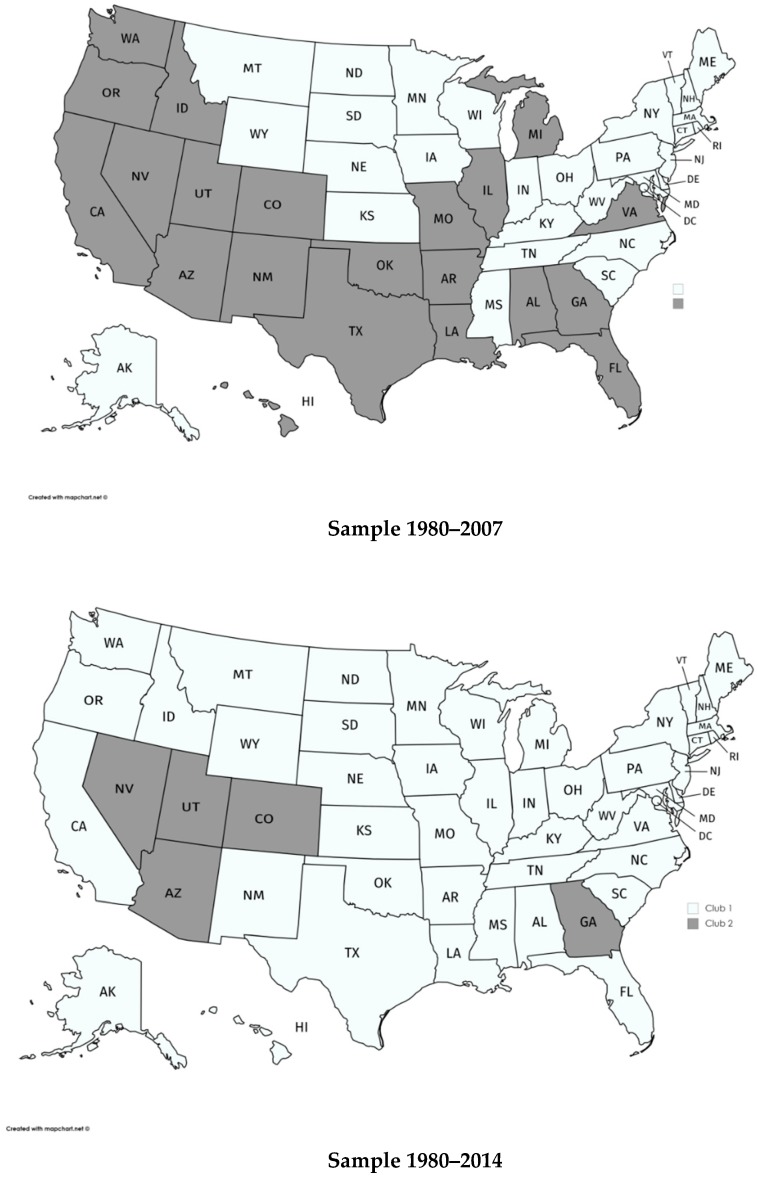
Estimated clubs for real per capita Personal Health Care Expenditure.

**Figure 2 ijerph-17-00554-f002:**
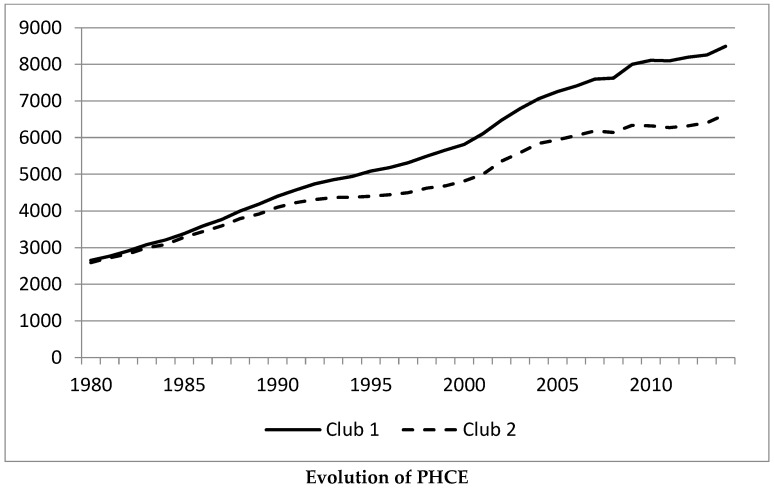
Average values of the PHCE of the states in Club 1 and Club 2 when the 1980–2014 sample is considered.

**Table 1 ijerph-17-00554-t001:** Testing for convergence.

	1980–2007	1980–2014
Panel I. Phillips-Sul test		
Personal Health Care	−0.58 (−45.2)	−0.78 (−66.0)
Panel II. Estimated Convergence clubs		
Club 1	AK, CT, DE, DC, IN, IA, KS, KY, ME, MD, MA, MN, MS, MT, NE, NH, NJ, NY, NC, ND, OH, PA, RI, SC, SD, TN, VT, WV, WI, WY	AL, AK, AR, CA, CT, DE, DC, FL, HI, ID, IL, IN, IA, KS, KY, LA, ME, MD, MA, MI, MN, MS, MO, MT, NE, NH, NJ, NM, NY, NC, ND, OH, OK, OR, PA, RI, SC, SD, TN, TX, VT, VA, WA, WV, WI, WY
Club 2	AL, AZ, AR, CA, CO, FL, GA, HI, ID, IL, LA, MI, MO, NV, NM, OK, OR, TX, UT, VA, WA	AZ, CO, GA, NV, UT

This table reports the results of the PS methodology for testing the null hypothesis of convergence. The different cells of Panel I present the value of the estimator of the log-t parameter and, below it, in parentheses, the PS statistic. The distribution of this statistic asymptotically converges towards a standard N(0, 1) distribution. So, we should use the −1.65 one-side critical value to reject the null hypothesis of convergence. Panel II includes the estimated convergence clubs, which have been obtained using the clustering algorithm designed in Phillips and Sul [10]. In all the cases, the Hodrick-Prescott filter has been employed, with the smoothing parameter being equal to 400. The different states are represented by their corresponding two-letter postal abbreviations.

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
