# Peer review of "Does the Great Recession Contribute to the Convergence of Health Care Expenditures in the US States?"

_ijerph, 2020, doi:10.3390/ijerph17020554_

Round 1

Reviewer 1 Report

To Authors,

I really enjoyed reviewing your manuscript. I have few concerns to improve your manuscript before publication.

Please add more recent literature reviews related with your topics to be up to date.

Great recession in the US should be effect on uncompensated care for patients.

From your data, is there any evidence for this effect? If so, please elaborate this for future readers.

In page 2 on line 74, the authors used the -1.65 one-side critical value to reject the null hypo. Can you please provide your rationale to use this criteria?

Please explain more about PS test for future readers.

Otherwise, well-written paper. 

Author Response

Referee #1

Thank you very much for your comments and suggestions. We have tried to include them into the new version.

We below provide a detailed answer to your questions

Point 1. Please add more recent literature reviews related with your topics to be up to date.

Response.

We have added the references [8], [9] and [13]. We have been not able to find more references related to the convergence analysis in the US.

Point 2. Great recession in the US should be effect on uncompensated care for patients. From your data, is there any evidence for this effect? If so, please elaborate this for future readers .

Response.

The database employed does not allow to study the effect on uncompensated care for patients. This would of course be a very interesting point to analyze, but we should need data which are not available (up to the best of our knowledge).

Point 3. In page 2 on line 74, the authors used the -1.65 one-side critical value to reject the null hypo. Can you please provide your rationale to use this criteria? Please explain more about PS test for future readers.

Response.

The PS statistic asymptotically follows a standard N(0,1) distribution. If we take into account that the null hypothesis of convergence is only rejected whenever the parameter b in (A3) takes negative values, then we focus on the left part of the distribution and use the -1.65 critical value. Following your suggestion, we have extended the explanation of the PS statistics in a separated Appendix, where we have tried to clarify this point.

Reviewer 2 Report

The Authors analyze a very relevant issue. They argue that "during the crisis the  decline in US health spending has been quite heterogeneous" and that:"some recent papers have analyzed  disparities, putting special emphasis on the convergence of the US regional health expenditure." (p.1)

They analyze personal Health Care data from 1980 to 2014  from the Center of Medicare and Medicaid Services (CMS) for the 50 US states plus the ones of the District of Columbia. Health care expenditure is measured by State of Provider.

The paper is novel and  its is worth publishing after some review.

- Page 3 you mention "Sample 1980-2017" , why?

In the text and abstract I understand that you analyzed data up to 2014.

as in page 2 I found that "We focus on State of Provider health data and our sample covers 1980-2014. Therefore, our results can be understood as an extension of those of Panopoulu and Pantelidis [7] for the post-Great Recession period."

This raises another question. In order to better understand the dynamics of the post-crisis you would need to add some more years  to the panel data, so can you please explain why you stopped at 2014? In any case I suggest to test further by using data up to 2017 and check if  you get the same results, otherwise to update the results obtained with up to 2014 data.

- Page 5 "In particular, it would be interesting to extend our study to include the analysis of health outcomes (infant mortality rate or life expectancy, for instance), an aspect that is left for future studies."

This study contributes to the literature by applying a dynamic economic growth model proposed by Phillips & Sul,  but does not follow  with a panel analysis on factors determining the growth of health expenditure in the US states as other recent studies for other inter country comparison do like, e.g.,  Nghiem and  Connelly,  Health Econ Rev. 2017; 7: 29.doi: 10.1186/s13561-017-0164-4, Convergence and determinants of health expenditures in OECD countries.  The determinants are several, not only health outcomes, but also GDP for example. Moreover, determinants of health care spending vary across States. Please  add this reference and justify why you left the analysis of the determinants for future studies as it seems to be an important complement of this analysis. 

 - I suggest to add an Appendix with with a synthesis and a discussion on the methods used, comparing the methodological approach (pros and cons) chosen for the estimation of convergence with other methods available from the literature (see also Nghiem and Connelly, 2017) as this would improve the paper. Also, please discuss at least what could have determined the different behaviour of the clubs.

Author Response

Reply to referee #2

Thank you very much for your comments and suggestions. We have tried to include them into the new version.

We below provide a detailed answer to your questions (in red)

Point 1. – “Page 3 you mention "Sample 1980-2017" , why?

In the text and abstract I understand that you analyzed data up to 2014. As in page 2 I found that "We focus on State of Provider health data and our sample covers 1980-2014. Therefore, our results can be understood as an extension of those of Panopoulu and Pantelidis [7] for the post-Great Recession period."

This raises another question. In order to better understand the dynamics of the post-crisis you would need to add some more years to the panel data, so can you please explain why you stopped at 2014? In any case I suggest to test further by using data up to 2017 and check if  you get the same results, otherwise to update the results obtained with up to 2014 data.”

Response. The sample employed across the paper is 1980-2014. We have corrected it into the new version. We agree the referee that increasing the sample size would offer a better scope of the effect of the crisis on health care expenditure disparities, but the database only contains information up to 2014.

Point 2. “- Page 5 "In particular, it would be interesting to extend our study to include the analysis of health outcomes (infant mortality rate or life expectancy, for instance), an aspect that is left for future studies."

This study contributes to the literature by applying a dynamic economic growth model proposed by Phillips & Sul,  but does not follow  with a panel analysis on factors determining the growth of health expenditure in the US states as other recent studies for other inter country comparison do like, e.g.,  Nghiem and  Connelly,  Health Econ Rev. 2017; 7: 29.doi: 10.1186/s13561-017-0164-4, Convergence and determinants of health expenditures in OECD countries.  The determinants are several, not only health outcomes, but also GDP for example. Moreover, determinants of health care spending vary across States. Please  add this reference and justify why you left the analysis of the determinants for future studies as it seems to be an important complement of this analysis.”

Response. We agree with the referee again in the sense that it is of great interest to analyze the determinants of the health care expenditure, especially so far changes in the GDP elasticity is concerned. In fact, we are working in this issue at this moment. We have decided to omit this part from this present paper given that we consider that it is necessary to employ a quite different econometric methodology (unit roots with changes in the trend function plus the use of the methodologies proposed in either Bai and Perron, 1998, or Perron and Qu, 2006). Then, we have preferred to separate both analyses. Furthermore, we posed this paper as a brief note and the inclusion of this new analysis would exceed the length limits for this type of papers. In any event, we have added the reference (thank you very much because we did not know it) and we explain in the text that we leave the analyze of the determinants for future research.

Bai, J., & Perron, P. (1998). Estimating and testing linear models with multiple structural changes. Econometrica, 47-78.

Perron, P., & Qu, Z. (2006). Estimating restricted structural change models. Journal of Econometrics134(2), 373-399.

Point 3 – “I suggest to add an Appendix with a synthesis and a discussion on the methods used, comparing the methodological approach (pros and cons) chosen for the estimation of convergence with other methods available from the literature (see also Nghiem and Connelly, 2017) as this would improve the paper. Also, please discuss at least what could have determined the different behaviour of the clubs.”

Response. Thank you very much for this suggestion. We have included an Appendix where the PS methodology is presented, including a brief discussion of its pros and cons.

We have added a small discussion at the end of Section 3 (lines 100-105) related to the forces that may have created the estimated clubs.